# Metabolic Flexibility of the Heart: The Role of Fatty Acid Metabolism in Health, Heart Failure, and Cardiometabolic Diseases

**DOI:** 10.3390/ijms25021211

**Published:** 2024-01-19

**Authors:** Virginia Actis Dato, Stephan Lange, Yoshitake Cho

**Affiliations:** 1Division of Cardiovascular Medicine, Department of Medicine, University of California San Diego, La Jolla, CA 92093, USA; vactisdato@health.ucsd.edu (V.A.D.); slange@health.ucsd.edu (S.L.); 2Department of Biomedicine, Aarhus University, DK 8000 Aarhus, Denmark; 3Steno Diabetes Center Aarhus, Aarhus University Hospital, DK 8200 Aarhus, Denmark

**Keywords:** fatty acid metabolism, heart failure, mitochondrial oxidative metabolism, metabolic flexibility, cardiometabolic diseases

## Abstract

This comprehensive review explores the critical role of fatty acid (FA) metabolism in cardiac diseases, particularly heart failure (HF), and the implications for therapeutic strategies. The heart’s reliance on ATP, primarily sourced from mitochondrial oxidative metabolism, underscores the significance of metabolic flexibility, with fatty acid oxidation (FAO) being a dominant source. In HF, metabolic shifts occur with an altered FA uptake and FAO, impacting mitochondrial function and contributing to disease progression. Conditions like obesity and diabetes also lead to metabolic disturbances, resulting in cardiomyopathy marked by an over-reliance on FAO, mitochondrial dysfunction, and lipotoxicity. Therapeutic approaches targeting FA metabolism in cardiac diseases have evolved, focusing on inhibiting or stimulating FAO to optimize cardiac energetics. Strategies include using CPT1A inhibitors, using PPARα agonists, and enhancing mitochondrial biogenesis and function. However, the effectiveness varies, reflecting the complexity of metabolic remodeling in HF. Hence, treatment strategies should be individualized, considering that cardiac energy metabolism is intricate and tightly regulated. The therapeutic aim is to optimize overall metabolic function, recognizing the pivotal role of FAs and the need for further research to develop effective therapies, with promising new approaches targeting mitochondrial oxidative metabolism and FAO that improve cardiac function.

## 1. Introduction

The human heart is an energetic omnivore, and it depends on the use and generation of ATP at a high rate to sustain its basal metabolism and contractile function. In cardiomyocytes, ATP is consumed in several processes, such as the pumping of ions across the membrane via the SERCA ATPase, which helps in maintaining calcium homeostasis, and in actin–myosin cross-bridge cycling, where it provides the energy required for the contraction and relaxation of heart muscle cells. The high cardiac metabolic flexibility allows it to switch between different energetic substrates depending on the situation [1]. In a healthy heart, mitochondrial oxidative metabolism is the primary energy source [2]. Except for in the postprandial state, about 80% of cardiac ATP is provided by fatty acid oxidation (FAO) in the mitochondria [3]. The remainder is derived from the oxidation of glucose and lactate, as well as small amounts of ketone bodies and amino acids [2,3]. All these cardiac fuels must be acquired continuously from the blood due to the low ability of the heart to store these energy substrates intracellularly [4,5].

## 2. The Mechanism of Fatty Acid Metabolism and Regulation in the Heart

During cardiac development, the metabolic fuel preference shifts from glycolysis in the fetal stage to FAO in the adult heart. This change is crucial, as FAs provide a more efficient energy source and stimulate mitochondrial biogenesis, thereby ensuring that the heart can meet the increased energy demands of the growing organism [2,3]. FAO involves three main steps: FA uptake into the cytosol; transport across the mitochondrial membrane; and oxidation in the mitochondrial matrix, which is the innermost compartment of the mitochondria [6]. Exogenous FAs are delivered to the heart either bound to albumin in the blood or hydrolyzed from the triacylglycerols (TGs) contained in chylomicrons and very-low-density lipoproteins (VLDLs) following hydrolysis by lipoprotein lipase [2,4]. The rate of FA uptake is influenced by the plasma levels of free FAs, which can be raised by heart failure (HF), ischemia, obesity, or diabetes [1,4,6,7]. FAs move across cell membranes through a process involving passive “flip-flop” translocation, as well as active transport via FA translocase (FAT/CD36) and cytoplasmic fatty acid binding proteins (FABPs) [4,8].

Once inside the cell, FAs are converted to fatty acyl-CoA by the enzyme acyl-CoA synthetase-1 (ACSL1) [9]. Short- and medium-chain FAs can directly enter the mitochondria, while long-chain variants require the carnitine palmitoyltransferase system (CPT 1/2) [2,10]. Inside the mitochondria, fatty acyl-CoA undergoes β-oxidation to produce acetyl-CoA, which enters the tricarboxylic acid (TCA) cycle, flavin adenine dinucleotide (FADH_2_), and nicotinamide adenine dinucleotide (NADH) to generate ATP [11,12] (Figure 1). Very-long-chain FAs may also undergo β-oxidation in peroxisomes [2,4]. The regulation of FAO in the heart is a multifaceted process involving a network of enzymes, molecules, and regulatory mechanisms. This regulation includes circulating FA supply to the heart [1,2], cellular FA uptake [4,8], malonyl CoA-induced CPT-1 inhibition [13], the post-translational modification of FA oxidative enzymes [3,14,15], and the transcriptional regulation of FA oxidative enzyme expression [3,16,17].

It is well established that peroxisome proliferator-activated receptors (PPARs: PPARα, γ, and δ) and estrogen-related receptors (ERRs: ERRα, β, and γ) play crucial roles in regulating the expression of genes involved in FA metabolism [17,18]. While free FAs are ligands that activate PPARs, ERR activity is mainly modulated by co-regulators [19]. PPARs and ERRs control the transcription of genes essential for FA transport and β-oxidation, such as CD36, FATP, CPT1, ACOX, and MCAD. Beyond FA metabolism, PPARs and ERRs also regulate the genes involved in glucose metabolism, lipid synthesis, cell proliferation, and differentiation, such as GLUT4 (glucose uptake) and ApoA-I and -II (HDL lipoproteins and lipid transport) [17,18] (Figure 1). Thus, these data suggest that PPARs and ERRs govern gene networks for fuel metabolism in the heart. While PPARs and ERRs share overlapping functions, they also play distinct roles in cardiac metabolism. Supporting this notion, a loss of PPARs and ERRs can lead to metabolic imbalances and disorders. Hepatocyte PPARα deletion impaired FA catabolism, leading to hepatic lipid accumulation during fasting, insulin resistance, and cardiovascular disease [17,20,21]. Moreover, in a study of adipose-specific PPARα knockout mice, it was found that the loss of PPARα signaling in adipocytes caused increased cholesterol esters and the activation of sterol regulatory element-binding protein-1 (SREBP-1), leading to a shift in macrophage polarity and increased lipogenesis [22]. Recent studies show that cardiac-specific PPARα knockout mice show a reduced gene expression of FAO and accelerated pressure overload-induced cardiac dysfunction [23,24]. Cardiac-specific ERRα knockout mice showed cardiac dysfunction, mild glucose and insulin intolerance, and reduced ERα gene expression in skeletal muscle and white adipose tissues [25]. These results suggest that proper control of FA metabolism is critical to maintain cardiac development and function.

Peroxisome proliferator-activated receptor gamma coactivator 1 (PGC-1) family, particularly PGC-1α, interact with both PPARs and ERRs to enhance FA uptake and FAO [17,18]. It is well established that PGC-1 plays a vital role in mitochondrial biogenesis, essential for cellular energetics [26,27]. A loss of PGC-1α has been linked with obesity, cardiovascular disease, and hepatic steatosis [28]. However, the activation of PGC-1α stimulates mitochondrial biogenesis and promotes muscle remodeling to a more oxidative phenotype [29]. PGC-1α and β regulate comprehensive genetic programs, including the activation of FAO and oxidative phosphorylation [30]. The activity of PGC-1α is further modulated by its expression levels and post-translational modifications, such as phosphorylation, acetylation, and methylation [31]. Interestingly, the palmitoylation and S-acylation of cysteines with saturated FAs play an essential role in myocardial electrophysiology by regulating sodium and calcium transport [32]. Further studies are needed to elucidate the role of FA modification in physiological and pathophysiological conditions.

Critically, the intricate regulation of FA utilization can be disrupted by alterations in substrate availability and uptake [1]. Conditions like obesity, diabetes, and other metabolic disorders can impact cardiac metabolism and mitochondrial function, triggering a cascade of effects, including disturbances in calcium homeostasis, an increased production of reactive oxygen species (ROS), and even the initiation of pro-apoptotic pathways [33,34]. Consequently, the failing heart loses its metabolic flexibility and decreases its ability to produce ATP compared to a healthy heart [1,3].

## 3. Fatty Acid Metabolism in Diverse Diseases

Beyond the effect at the cardiac level, alterations in FA metabolism play a role in the pathophysiology of different diseases in humans and animals. In the obesity state, an excess FA intake leads to their increased storage in adipose tissues, disrupting normal metabolic processes, often resulting in insulin resistance, a hallmark of type 2 diabetes mellitus (T2DM). Insulin resistance impairs the ability of cells to use glucose effectively; this condition is usually accompanied by energy deficiency and metabolic disturbances. Insulin resistance is further exacerbated by the release of increased amounts of non-esterified FAs, pro-inflammatory cytokines, and other factors from adipose tissue in obese individuals that affect energy metabolism in the whole body [35,36,37].

Metabolic syndrome, a cluster of conditions that encompasses hypertension, hyperglycemia, excess body waist fat, and abnormal cholesterol levels, is also associated with disturbed FA metabolism. In this sense, it was described that hepatic FAO and ketogenesis drain FAs from blood and extrahepatic tissues, contributing significantly to decreasing fat mass accumulation and improving peripheral insulin sensitivity. Experiments in PPARα-null mice showed that circulating FA clearance resulted from an induction of hepatic lipoprotein lipase (LPL) activity and FA re-uptake from very-low-density lipoprotein (VLDL). This was associated with increased hepatic FAO, causing hypolipidemia, an anti-adiposity effect, and improved insulin sensitivity [38]. Regarding hepatic tissue, FA metabolism and regulation are not only involved in metabolic pathologies but are also associated with malignancy, prognosis, and immune phenotypes in hepatocellular carcinomas (HCCs). Several genes related to FA metabolism are closely associated with the development and progression of HCCs. These biomarkers play a role in the induction of HCCs and are potentially diagnostic tools for the early detection and clinical management of disease [39].

FAs are also involved in the pathogenesis of certain neurological disorders, such as Alzheimer’s disease. It was recently found that the brain critically depends on astrocytic FAO to maintain lipid homeostasis. Aberrant astrocytic FAO-induced lipid accumulation is followed by neurodegeneration, synaptic loss, neuroinflammation, demyelination, and cognitive impairment [40]. The dysregulation of FA metabolism can also contribute to chronic inflammation, a critical feature of various diseases, including rheumatoid arthritis and inflammatory bowel disease [41,42]. Disorders like muscular dystrophy are also associated with altered FA metabolism. In mice with muscular dystrophy, an initial inability to utilize FAs for energy via mitochondrial FAO resulted in the shunting of FAs into triacylglycerol as the disease progressed [43].

Finally, cardiovascular diseases (CVDs) encompass a range of conditions affecting the heart and blood vessels, and they are intricately linked with FA metabolism. This relationship is primarily observed in atherosclerosis, where altered FA metabolism leads to dyslipidemia, characterized by abnormal levels of lipids in the circulation. Elevated levels of low-density lipoprotein (LDL) cholesterol and triglycerides, along with reduced high-density lipoprotein (HDL) cholesterol, contribute to atherosclerosis [44]. Plaque formation in the arterial walls is directly influenced by circulating FA levels, leading to an increased risk of coronary artery disease, myocardial infarction, and stroke [45]. Mice treated with Betulin, an inhibitor of sterol regulatory element-binding proteins (SREBPs), showed reduced FA novo synthesis and atherosclerotic plaque formation. SREBPs were previously found to be involved in the biosynthesis of cholesterol, FAs, and triglycerides. These data suggest the direct contribution of FAs to the development of atherosclerosis [46]. In this sense, there is growing evidence demonstrating that FAs can influence the function of T cells, favoring their activation, proliferation, and polarization, thus contributing to the development and progression of atherosclerosis [47]. Moreover, it is well established that FAs can have pro-inflammatory effects via the Toll-like receptor 4 (TLR4)/NFκB pathway in macrophages [48]. Recent studies showed that the interaction between FAs and CD36 can integrate cell signaling and metabolic pathways through AMP-activated protein kinase (AMPK) and NFκB, thereby influencing immune cell differentiation and activation, and promoting foam cell formation [49]. The activation and polarization of T cells and macrophages in an inflammatory environment, together with a lipid-rich milieu in the vasculature, are critical for the development and progression of atheroma plaque.

In summary, the intricate relationship between FA metabolism and various health conditions underscores this metabolic pathway’s significance in humans and animals. The proper management of FA metabolism is essential for cardiovascular health, as well as for preventing and treating a wide array of diseases, including obesity, T2DM, and metabolic syndrome. These conditions are often interconnected, sharing common pathways influenced by FA metabolism. The balance of FAs is crucial in maintaining optimal cardiovascular function. At the same time, imbalances lead to cardiac energy deficiency and the development of atherosclerosis and other cardiovascular diseases as the main consequences. In the subsequent sections, this review delves into the leading role of FA metabolism, specifically in the heart, particularly under the influence of widespread cardiovascular pathologies, including heart failure and diabetic cardiomyopathy, offering a comprehensive analysis of its impact and implications in these prevalent conditions.

## 4. Fatty Acid Metabolism in Heart Failure with Reduced Ejection Fraction (HFrEF) and Preserved Ejection Fraction (HFpEF)

Heart failure (HF) is a severe medical condition characterized by the inability of the heart to pump sufficient blood to supply nutrients and oxygen to all body tissues. This leads to significant disabilities and high mortality rates [50]. One of the critical features of HF is compromised energy metabolism and metabolic imbalance [1,6]. Factors contributing to this phenomenon include impaired mitochondrial oxidative metabolism, changes in the preference for energy substrates, and decreased cardiac efficiency [1,6].

HF is primarily categorized into two types: heart failure with a reduced ejection fraction (HFrEF) and heart failure with a preserved ejection fraction (HFpEF), which is more prevalent [51]. The evidence regarding the uptake and utilization of FAs as energy fuel in the different models and stages of HF with comorbidities is inconsistent across all studies. HFrEF patients generally exhibit increased levels of plasma FAs, increased FA metabolites, and FAO. However, this does not correspond to increased FA levels within the myocardium itself [3,52,53,54]. In addition, in the HFrEF condition, there is an increase in ketone body oxidation, potentially leading to a reduction in FAO and glucose oxidation. Moreover, the oxidation of branched-chain amino acids is compromised in HFrEF, indicating a distinct metabolic impairment that could have further implications for cardiac energy dynamics and heart failure pathology [55].

Conversely, in HFpEF, the myocardium generally exhibits lower levels of FAs, ketones, and TCA metabolites than in HFrEF, suggesting fuel inflexibility in this syndrome [53]. These findings are dependent on the progression of HFpEF and the presence of comorbidities. HFpEF patients with aortic stenosis displayed an increased FA uptake, leading to decreased glucose utilization and an increased generation of ROS [56]. However, several other studies reported that myocardial FA uptake and the expression of FAO enzymes were unchanged [57] or even decreased in HFpEF patients [53,58,59]. As HFpEF progresses, FA uptake decreases, and there is a shift towards an increased utilization of ketone bodies and lactate, along with higher rates of proteolysis in different animal and human models [58,60,61]. Recently, a HFpEF mouse model showed global protein hyperacetylation in the heart; the enrichment of mitochondrial proteins involved in the TCA cycle; oxidative phosphorylation (OXPHOS) and FAO, which correlated with a reduced NAD+/NADH ratio; and impaired mitochondrial function [59,60]. Moreover, in a rat HFpEF model, metabolic syndrome affected cardiac protein expression, promoting the hyperacetylation of proteins regulating FA metabolism [62]. Finally, metabolomic analyses showed that metabolites are enriched in specific metabolic pathways in the hearts of HFpEF mice, including the biosynthesis of unsaturated FAs, lipids, amino sugar and nucleotide glucose, glycerophospholipid, arachidonic acid, and arginine and proline. In contrast, studies showed that enriched metabolites in HFrEF mice were mainly related to arginine and proline metabolism; glycine, serine, and threonine metabolism; pantothenate and CoA biosynthesis; glycerophospholipid metabolism; nicotinate and nicotinamide metabolism; and arachidonic acid metabolism [63]. These metabolic changes suggest distinct and HF-specific alterations to FA metabolism between HFpEF and HFrEF.

Related to the molecular mechanism of FAO, a decrease in the progression of HF may be attributed, in part, to reduced expressions of FA transporters [55,57,61,64]. Moreover, during the progression of HF, a reduced PPARα expression was reported, resulting in a decreased expression and activity of key FA transporter proteins like CPT1 or CPT2 [65,66]. Finally, in advanced HF stages, a metabolic mismatch was found between FA uptake and FAO [67,68]. While FAO decreases, FA uptake into the cytoplasm is unaltered, leading to intracellular lipid accumulation [68]. This lipid excess generates toxic lipid species, contributing to mitochondrial dysfunction, with increased mitochondrial fission and reduced fusion, apoptosis, and the progression of HF [69,70,71]. Additionally, lipid accumulation induces insulin resistance by affecting components of the insulin signaling cascade [72,73,74]. This evidence suggests that an oversupply of metabolic substrates could exacerbate cardiac dysfunction [74,75]. Despite a few discrepancies, all these findings generally converge to indicate that cardiac FA uptake into the mitochondria and FAO diminishes as HF progresses, regardless of the specific changes observed at each stage.

Recognizing and understanding these metabolic shifts in HF are crucial for devising targeted therapeutic approaches. Although FAO is a significant area of focus in this context, data discrepancies underscore the need for more research to gain a deeper understanding of the underlying mechanisms.

## 5. Fatty Acid Metabolism in Obesity and Diabetic Cardiomyopathy

Obesity and both types of diabetes, type 1 (T1DM) and type 2 (T2DM), have been associated with a greater risk of developing cardiomyopathy and HF, with a negative impact on the management of cardiovascular disease in the last decade [76,77]. A wide range of cellular and molecular mechanisms are involved in the etiology of cardiomyopathy developed under obesity and diabetes, including adipose tissue dysfunction, metabolic disturbances (insulin resistance, abnormal glucose and FA uptake, and lipotoxicity), altered mitochondrial calcium homeostasis, oxidative stress, systemic inflammation, autophagy, and mitophagy defects, as well as myocardial fibrosis, among others [78].

Focusing on cardiac FA metabolism during obesity and diabetes, there are significant shifts in cardiac metabolism, making the heart almost entirely reliant on FAO for ATP production. This dependence on FAO is particularly pronounced in T2DM, especially when combined with conditions like obesity and insulin resistance, which further exacerbate the reliance on FA uptake and FAO [79]. In T1DM, this shift towards FAO is primarily due to impaired insulin-dependent glucose uptake [3]. In this sense, some studies suggest that, by boosting mitochondrial FAO during obesity, cardiac dysfunction can be prevented [80]. However, transgenic mouse models in which cardiac FAO is blocked exhibited cardiac hypertrophy and accelerated impairment in ejection fraction in response to pressure overload, suggesting the importance of maintaining homeostasis, including balanced FAO [81]. However, although FAO is promoted during obesity and diabetes, its efficiency is diminished, leading to a mismatch between FA uptake and FAO, contributing to lipotoxicity [82,83,84]. Recently, a transgenic mouse model of cardiac lipotoxicity overexpressing ACSL1 (long-chain acyl-CoA synthetase 1) in cardiomyocytes suggested that excessive FA uptake leads to mitochondrial structural remodeling and increased mitochondrial fission. Observed molecular changes included the ubiquitination of AKAP121 (A-kinase anchor protein 121), a reduced phosphorylation of DRP1 (dynamin-related protein 1) at Ser637, and an altered proteolytic processing of OPA1 (optic atrophy-1). This was associated with increased palmitoyl-carnitine oxidation and ROS generation [84]. Moreover, excessive lipid supply facilitated DRP1 acetylation, which, in turn, increased its activity and mitochondrial translocation, resulting in cardiomyocyte dysfunction and death in mice fed a high-fat diet (HFD) with signs of obesity and T2DM [82].

The loss of metabolic flexibility during obesity or diabetes is also linked to decreased glucose oxidation, with concomitant cardiac hypertrophy and dysfunction in several studies [3,33,85]. This could be related to a reduced insulin response and transport of glucose transporter type 4 (GLUT4) for glucose uptake due to a lower availability or activity of essential proteins involved in insulin signaling, such as the insulin receptor (IR) and insulin receptor substrate-1 (IRS-1) [73,74,86]. Moreover, in T2DM, hyperglycemia is another primary driver of cardiac dysfunction, leading to oxidative stress and promoting the formation of advanced glycation end products, inflammation, cell death, and post-translational protein modifications in the diabetic heart [3,87]. It was recently found that glycation end products promoted the glycation damage of RyR2 (ryanodine receptor 2), contributing to calcium leaks and mitochondrial damage in the myocardium of mice [88].

In conclusion, under obesity and diabetes, the heart’s reliance on FAO for energy production contributes to diabetic cardiomyopathy. This metabolic shift leads to inefficiencies in energy conversion, lipotoxicity, and cardiac dysfunction. Understanding these complex metabolic pathways could provide new avenues for therapeutic intervention in diabetes-associated cardiac diseases.

## 6. Modulation of Fatty Acid Metabolism Regulation as Metabolic Therapy for Treating Cardiac Diseases

FA utilization has been the main target of research on therapy for cardiac diseases for decades, but the results are controversial. Different studies showed a relationship between increased FAO and cardiac dysfunction [3,58,85].

### 6.1. Inhibition of FAO

One strategy tested in hearts involves inhibiting FAO through drugs like trimetazidine (TMZ), etomoxir, and perhexiline, which have shown promise in animal models and human studies, primarily for conditions like ischemia–reperfusion and chronic HF [14,89,90]. This approach inhibits FAO, which may improve oxygen efficiency and restore coupling between glycolysis and glucose oxidation [91]. Based on this, one of the most widely used therapies is the CPT1A inhibitor TMZ, which has a cardioprotective effect that attenuates HF by improving myocardial metabolism via AMPK [92,93]. Furthermore, TMZ had the advantage of not modifying the heart rate or blood pressure, making this therapeutic agent an attractive clinical option [94]. An alternative approach is the use of PPARα agonists (fibrates), which reduce the myocardial supply of FAs by increasing their utilization in extracardiac tissues [95]. Treatment with fibrates had cardioprotective effects during ischemia–reperfusion injury, but clinical trials showed negative results on the risk of new-onset HF [96]. Another widely used strategy is the inhibition of malonyl CoA decarboxylase (MCD), which regulates malonyl CoA, an endogenous inhibitor of FA uptake in the heart. The inhibition of MCD in a rat model of HF led to an increase in myocardial malonyl CoA levels and a decrease in cardiac FAO rates, thus preventing HF development [14]. Recently, sodium–glucose co-transporter 2 (SGLT2) inhibitors, primarily known for their role in managing diabetes by promoting urinary glucose excretion, have been found to exhibit additional cardiovascular benefits [97]. In murine and porcine models, it was recently found that SGLT2 inhibition increases cardiac ketone oxidation [98,99], ameliorating cardiac dysfunction by providing the heart with a more oxygen-efficient substrate than FAO [100,101]. This alteration in substrate utilization is thought to alleviate metabolic stress and inflammation in the heart, potentially contributing to improved cardiac function [102] (Table 1). However, despite all the evidence, the inhibition of FAO strategies may not be effective in HF, where FAO is already impaired, and glucose oxidation is decreased and redirected toward biosynthetic pathways [53,103,104]. Thus, therapeutic approaches in HF might need to focus on restoring balance in all contributing metabolic pathways rather than solely reducing FAO in the heart.

### 6.2. Stimulation of FAO

Growing evidence is related to the stimulation of FAO as a strategy to improve cardiac energetics and contractility in the failing heart [3,105]. Studies have shown that increased FAO can attenuate fibrosis and ventricular dysfunction in hypertension, myocardial infarction, and genetic cardiomyopathy [106,107]. In this sense, there is a growing interest in the nuclear transcription factor retinoid X receptor alpha (RXRα) as a regulator of energy metabolism in cardiovascular diseases. Neocryptotanshinone (NCTS), with a high binding affinity for RXRα, was recently found to protect against HF post-acute myocardial infarction by improving cardiac dysfunction and attenuating cellular hypoxic injury. NCTS enhanced CD36 and CPT1a expression, increasing FA uptake and FAO. Moreover, NCTS enhanced mitochondrial transcription factor A (TFAM) levels and promoted mitochondrial biogenesis [108]. Another strategy of broad current interest is to increase FAO by improving mitochondrial biogenesis and function. Recently, it was found that increasing FAO by inducing acetyl coenzyme A carboxylase 2 (ACC2) deletion prevents obesity-induced cardiomyopathy, partly by maintaining mitochondria function through regulating parkin-mediated mitophagy [104] (Table 1).

The most recent studies suggest therapeutic concepts that aim to recover metabolic flexibility in the heart. These approaches recognize the complexity of metabolic remodeling in HF and suggest that strategies should optimize overall metabolic function rather than solely targeting FAO or glucose utilization [53,109].

## 7. Conclusions

In the realm of cardiac diseases like HF, the strategies to modify FAO are multifaceted and continually evolving. Although FAO inhibition as a therapeutic strategy is intriguing, researchers are also exploring methods to stimulate FAO and optimize overall metabolic function. Given that cardiac energy metabolism is intricate and tightly regulated, with FAs playing a pivotal role in ATP generation, understanding these metabolic pathways in pathophysiological conditions is critical for developing effective therapies for cardiac diseases. Considering all the evidence commented on in this review, treatment strategies should be tailored to individual patients and the underlying metabolic abnormalities in their hearts.

HF is a complex condition characterized by several abnormalities in cardiac lipid metabolism, including impaired FAO and TGs dynamics, along with the accumulation of lipid signaling molecules like diacylglycerol (DAG) and ceramide. Changes are also observed in the metabolism of ketone bodies, glucose, and amino acids. A mismatch between FA uptake and FAO leads to lipid accumulation, generating lipotoxicity, ROS production, and cell death. These alterations may reflect the myocardial intrinsic mechanisms used to adapt to the cardiac pathology, but they ultimately lead to ATP depletion and contractile dysfunction (Figure 2).

Research has focused on changes in myocardial FA and glucose oxidation as potential treatments for HF. Recent findings on anabolic and regulatory processes have expanded our understanding of metabolic remodeling in HF, highlighting new potential therapeutic targets. Moreover, cardiac energy metabolism draws energy from several circulating substrates, with the mitochondria playing a central role in ATP production. In the failing heart, multiple levels of this machinery are disrupted, including substrate utilization, mitochondrial oxidative metabolism, redox regulation, and energy transfer between the mitochondria and the cytosol. While significant progress has been made in understanding metabolic derangements in heart disease, further studies are needed to translate this knowledge into practical therapeutic approaches in specific disease conditions.

From our perspective, the core issue in HF is an energy deficiency, primarily due to a decreased mitochondrial oxidative capacity, increased glycolysis, and alterations in FAO and glucose metabolism. The therapeutic approach of targeting mitochondrial oxidative metabolism and FAO to enhance cardiac efficiency, mitigate energy deficits, and improve cardiac function in HF seems particularly promising.

## Figures and Tables

**Figure 1 ijms-25-01211-f001:**
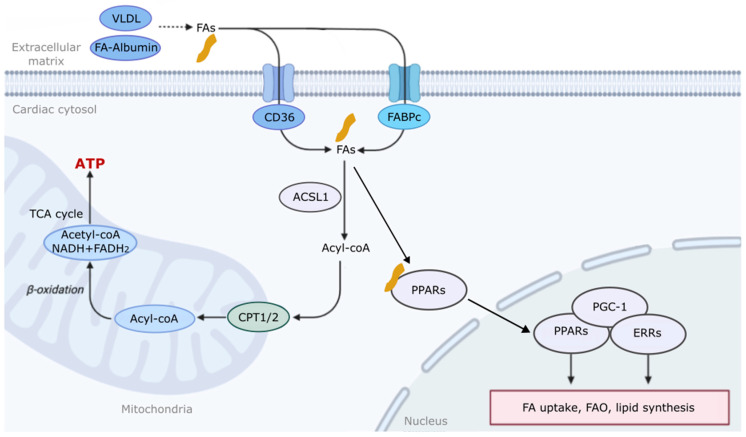
Overview of fatty acid metabolism in the heart. Exogenous fatty acids (FAs) are delivered to cardiomyocytes via albumin in the blood or hydrolyzed from triacylglycerols contained in very-low-density lipoproteins (VLDLs). FAs move across cell membranes through “flip-flop” passive translocation, as well as by active transport via FA translocase (FAT/CD36) and cytoplasmic fatty acid binding proteins (FABPs). FAs are converted to Acyl-CoA by the enzyme acyl-CoA synthetase-1 (ACSL1). Short- and medium-chain FAs can directly enter the mitochondria, while long-chain variants require the carnitine palmitoyltransferase system (CPT 1/2). Inside the mitochondria, fatty acyl-CoA undergoes β-oxidation to produce acetyl-CoA, which enters the tricarboxylic acid (TCA) cycle, flavin adenine dinucleotide (FADH_2_), and nicotinamide adenine dinucleotide (NADH) to generate adenosine triphosphate (ATP). Free FAs also activate peroxisome proliferator-activated receptors (PPARs). PPARs, peroxisome proliferator-activated receptor gamma coactivator 1 (PGC-1), and estrogen-related receptors (ERRs) control the transcription of genes essential for FA transport and β-oxidation and lipid synthesis in the nucleus.

**Figure 2 ijms-25-01211-f002:**
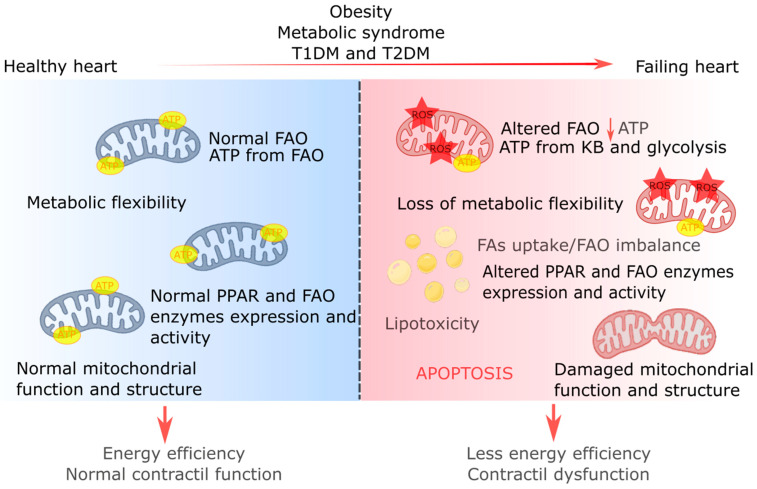
Overview of energy metabolism and cardiac remodeling in a healthy heart and a failing heart. A healthy heart has an energy metabolism mainly dependent on fatty acid oxidation (FAO) to produce adenosine triphosphate (ATP). Fatty acid (FA) metabolism is closely regulated by different enzymes and transcription factors, such as peroxisome proliferator-activated receptors (PPARs), which, together with a normal mitochondrial function and structure, ensure energy efficiency and normal contractile function of the heart. Obesity, metabolic syndrome, and diabetes mellitus types 1 and 2 (T1DM and T2DM, respectively) have consequences at the cardiac level, generating a failing heart. This condition is associated with altered FAO and lower ATP production. The loss of metabolic flexibility causes the heart to produce ATP mainly from the metabolism of ketone bodies (KBs) and glycolysis. There is also an imbalance between the uptake of FAs into the mitochondria and FAO, leading to lipotoxicity and the production of reactive oxygen species (ROS). Moreover, in the failing heart, the expression and function of PPARs and FAO are altered. Finally, damage to mitochondrial function and structure also occurs. All these alterations together can potentially lead to cell death through apoptosis. As a result, the failing heart has less energy production and contractile dysfunction.

**Table 1 ijms-25-01211-t001:** Fatty acid oxidation metabolic modulators for cardiac diseases.

Metabolic Modulators	Metabolic Mechanism Affected	Cardiac Condition
CPT1 inhibitors Etomoxir Perhexiline	↓ FAO ↑ Glucose oxidation	Ischemia–reperfusion injury Chronic HF
Trimetazidine (TMZ)	↓ FAO ↑ AMPK signaling	Pressure overload HF
MCD inhibitors	↑ Malonyl CoA levels ↓ FA uptake ↓ FAO	HF Post-myocardial infarction Ischemia–reperfusion injury
ACC2 inhibitors	↑ FA uptake ↑ FAO ↑ Parkin-mediated mitophagy ↑ Mitochondrial function	Obesity-induced cardiomyopathy
PPARα agonists Fibrates	↑ FA uptake ↑ FAO ↓ Triglycerides ↑ Insulin sensitivity	Ischemia–reperfusion injury T2DM with CVD
SGLT2 inhibitors	↑ Ketone body oxidation ↑ Glucose metabolism ↓ FAO	T2DM with CVD T2DM with HFrEF T2DM with HFpEF
RXRα activators Neocryptotanshinone (NCTS)	↑ CD36 and CPT1 expression ↑ FA uptake ↑ FAO ↑ Mitochondrial biogenesis	HF Post-myocardial infarction

Abbreviations: CPT1, carnitine palmitoyltransferase 1; FAO, fatty acid oxidation; HF, heart failure; AMPK, AMP-activated protein kinase; MCD, malonyl CoA decarboxylase; FAs, fatty acids; ACC2, acetyl coenzyme A carboxylase 2; PPAR, peroxisome proliferator-activated receptor; SGLT2, sodium–glucose co-transporter 2; KB, ketone body; T2DM, type 2 diabetes mellitus; CVD, cardiovascular disease; HFrEF, heart failure with a reduced ejection fraction; HFpEF, heart failure with a preserved ejection fraction; RXRα, retinoid X receptor alpha; CD36, cluster of differentiation 36. ↑, Increased; ↓, Decreased.

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
