# Peer review of "Metabolic Flexibility of the Heart: The Role of Fatty Acid Metabolism in Health, Heart Failure, and Cardiometabolic Diseases"

_ijms, 2024, doi:10.3390/ijms25021211_

Round 1

Reviewer 1 Report

Comments and Suggestions for Authors

The draft is well-organized and comprehensive in terms of a review in Fatty Acid metabolism. I find it easy to read and structurally reasonable. I would recommend the authors clearly indicate the citations for the summarized figures for a better illustration of the mechanisms. 

The enclosed manuscript drafted by Dato et al. intends to extract and discuss the sophisticated mechanism of fatty acid metabolism in heart functions as well as pathogenic roles. Particularly, the authors highlighted the involvement of fatty acids oxidation in the mitochondria-associated pathogenesis. Given the facts that many potential compounds, such as PPARalpha agonists, are developed to revitalize the mitochondrial functions, the authors strategically implied the potential druggable targets along this fatty acid metabolic cascade.

Fundamentally, the authors provided an overview of the fatty acid metabolic pathways in a cardiac orientation. The two critical signals, CPT1/2 and PPAR-PGC1, were clearly illustrated with proper citations in the texts. The comprehensive description indeed provides a solid background and rationale of the intentions of targeting both pathways. The format might be slightly shifted due to editing issues so the subheading for the 3rd paragraph *FA metabolism in diverse diseases” can be easily missed out. The authors spent some contexts to elaborate the fatty acid metabolism in various cardiac dysfunctions and diseases, followed by a summary of all putative therapeutic targets via FA metabolic signals. A schematic illustrate concluded the concept of this review article indicating the therapeutic outcomes to the cardiac function regarding metabolic disorders can be remodeled. Generally speaking, this review is well-organized and comprehensively discussed the rationale, medical intentions, and proposals to the therapeutic intentions. It would be better if the authors can refer the information in the schemes to respective references, and a few additional tables to summarize the current progress of using the mentioned inhibitors/agonists for the therapeutic purposes.

Comments on the Quality of English Language

The English was well written, but minor rephrases may be required for a better reading experience. 

Reviewer 2 Report

Comments and Suggestions for Authors

Thank you for the opportunity to review this paper on metabolic flexibility in the heart. This review provides a very detailed description of the role of fatty acid metabolism and is useful for readers in this area. The paper would benefit further if the authors would add commentary on the following points

Should the authors not consider the association between sodium-glucose co-transporter (SGLT)1 inhibition and cardiac function?

Should the author not consider the association between natriuretic peptide (NP) and energy metabolism? (for example, NP handicap)

Within Fig. 2

Apoptosis is mentioned in the figure, but there is little explanation in the text. The authors should provide additional explanation.

The reader has difficulty understanding what KB is in the diagram.

What does the background pattern represent? Shouldn't it be a plain background?

6.Conclusion→7.Conclusion (P8L351)

Round 2

Reviewer 2 Report

Comments and Suggestions for Authors

OK.